METHODS AND RESOURCES

# Organelle landscape analysis using a multiparametric particle-based method

**Yoshitaka Kurikawa‡, Ikuko Koyama-Honda‡\*, Norito Tamura¤a, Seiichi Koike¤b, Noboru Mizushima** ⓘ \*

Department of Biochemistry and Molecular Biology, Graduate School of Medicine, The University of Tokyo, Tokyo, Japan

¤a Current address: Directors' Research Unit, European Molecular Biology Laboratory, Heidelberg, Germany
¤b Current address: Department of Life Sciences and Bioengineering, Graduate School of Science and Engineering, University of Toyama, Toyama, Japan
‡ These authors share first authorship on this work.
\* honikuko@m.u-tokyo.ac.jp (IK-H); nmizu@m.u-tokyo.ac.jp (NM)

**Data Availability Statement:** All relevant data are within the paper and its Supporting Information files.

**Funding:** This work was supported by JST ERATO Grant Number JPMJER1702 to N.M. from Japan

## Abstract

Organelles have unique structures and molecular compositions for their functions and have been classified accordingly. However, many organelles are heterogeneous and in the process of maturation and differentiation. Because traditional methods have a limited number of parameters and spatial resolution, they struggle to capture the heterogeneous landscapes of organelles. Here, we present a method for multiparametric particle-based analysis of organelles. After disrupting cells, fluorescence microscopy images of organelle particles labeled with 6 to 8 different organelle markers were obtained, and their multidimensional data were represented in two-dimensional uniform manifold approximation and projection (UMAP) spaces. This method enabled visualization of landscapes of 7 major organelles as well as the transitional states of endocytic organelles directed to the recycling and degradation pathways. Furthermore, endoplasmic reticulum–mitochondria contact sites were detected in these maps. Our proposed method successfully detects a wide array of organelles simultaneously, enabling the analysis of heterogeneous organelle landscapes.

## Introduction

Organelles separated by biological membranes play vital roles in cellular processes [1]. Studies on the morphology, composition, and temporal changes of organelles have improved our understanding of the function of each organelle. Organelles are comprised of heterogeneous populations involved in various processes, including division, fusion, organelle-to-organelle contact, and maturation [2]. Even minor components and intermediate structures can influence cellular functions [3]. To understand such heterogeneity of organelles, comprehensive yet simple methods are necessary.

Such methods require a sufficient number of parameters, high resolution, and unbiased sampling, but the methods commonly used in cell biology are inadequate. For example, biochemical methods such as immunoisolation and density gradient-based fractionation can analyze only a limited number of organelles at once, with unavoidable contamination between

Science and Technology Agency (https://www.jst.go.jp/EN/), JSPS KAKENHI Grand Numbers 22H04919 to N.M. and 22K06217 to I.K.-H. from Japan Society for the Promotion of Science (https://www.jsps.go.jp/english/index.html). The funders had no role in study design, data collection and analysis, decision to publish, or preparation of the manuscript.

**Abbreviations:** BIC, Bayesian information criterion; CLEM, correlative light and electron microscopy; DMEM, Dulbecco's Modified Eagle Medium; EGF, epidermal growth factor; ER, endoplasmic reticulum; HEK, human embryonic kidney; NHS, *N*-hydroxy-succinimidyl; PCA, principal component analysis; UMAP, uniform manifold approximation and projection.

fractions. Fluorescence microscopy can be used to observe multiple labeled organelles in cells, but the high density of cellular components makes it difficult to distinguish whether they are in contact or merely in close proximity [4,5]. Although multicolor imaging has been attempted with superresolution microscopy, it only partially resolves the issue of resolution [6–9]. Electron microscopy, particularly three-dimensional volume electron microscopy, can provide more detailed information at high resolution but generally has a limited field of view except for the recently developed high-throughput electron microscopy [4,10,11]. Recently, more comprehensive methods, such as machine learning–based multiple organelle segmentation at the whole cell level, have been developed; however, these methods require special equipment and skills, and the obtained data are difficult to interpret [12–14]. Furthermore, microscopy is generally limited in that only a small number of cells can be observed, leading to potentially biased data. Methods have been proposed to extract organelles as particles from cells and analyze them by flow cytometry [15–22], which may improve the spatial resolution and allow for more unbiased detection of organelles. However, most of these methods remain limited to analyses of a single type or a few types of organelle because the sensitivity for detecting multicolor fluorescence of small particles is insufficient. Although mass cytometry has been used for the detection of mitochondria, lysosomes, and autophagosomes, mass cytometers are not widely available [23]. Therefore, there is a trade-off between observing multiple organelles with sufficient resolution and unbiased sampling of organelles.

To overcome these limitations, in this study, we developed a simple multiparametric particle-based method. In our proposed method, isolated organelle particles labeled with multiple markers are analyzed by fluorescence microscopy using spectral imaging [5]. The obtained multidimensional data are then visualized as organelle landscape maps using dimension reduction techniques. Using these data, we were also able to successfully detect the endoplasmic reticulum (ER)–mitochondria contact sites and analyze the transitional states during endosome maturation. This technique is useful for elucidating the intracellular organelle landscape.

## Results

### Development of a method for analyzing organelle particles

To detect multiple organelles with high resolution, we isolated organelle particles from cells and performed multicolor imaging. HeLa cells expressing markers for the ER (mTagBFP2 (BFP)–SEC61B) [24], mitochondria (GFP–OMP25 and SNAP–OMP25) [25], and the Golgi (Venus–GS27) [26,27] were used. Early endosomes were labeled by 5-min incubation with Alexa Fluor 647 (Alexa647)-conjugated epidermal growth factor (EGF) (hereafter, Alexa647–EGF) at 37°C. The plasma membrane was labeled by incubating cells for 15 min with Alexa405-conjugated *N*-hydroxy-succinimidyl (NHS) ester (hereafter, Alexa405–NHS) at 4°C just before cell homogenization. After homogenization by gentle sonication, peroxisomes and lysosomes were stained with Alexa594-conjugated anti-PMP70 antibody and Alexa680-conjugated anti-LAMP1 antibody, respectively. Thus, the organelle particles were labeled with eight colors in total (Figs 1A and S1A). Correlative light and electron microscopy (CLEM) confirmed that membranous structures positive for the markers of these organelles, such as the mitochondria and ER, were contained in these samples (Fig 1B).

While multicolored particles can be detected by either flow cytometry or fluorescence microscopy, we chose fluorescence microscopy owing to its higher sensitivity for small particles (S1B Fig). Using a confocal fluorescence microscope equipped with spectrometers (S1B Fig), lambda scanning was performed between wavelengths of 411 nm and 736 nm (S1C Fig). Single-color organelle particles labeled with different fluorescent dyes were used to acquire spectral data, and linear unmixing was performed to obtain 8-color fluorescence images (Fig 1C). To

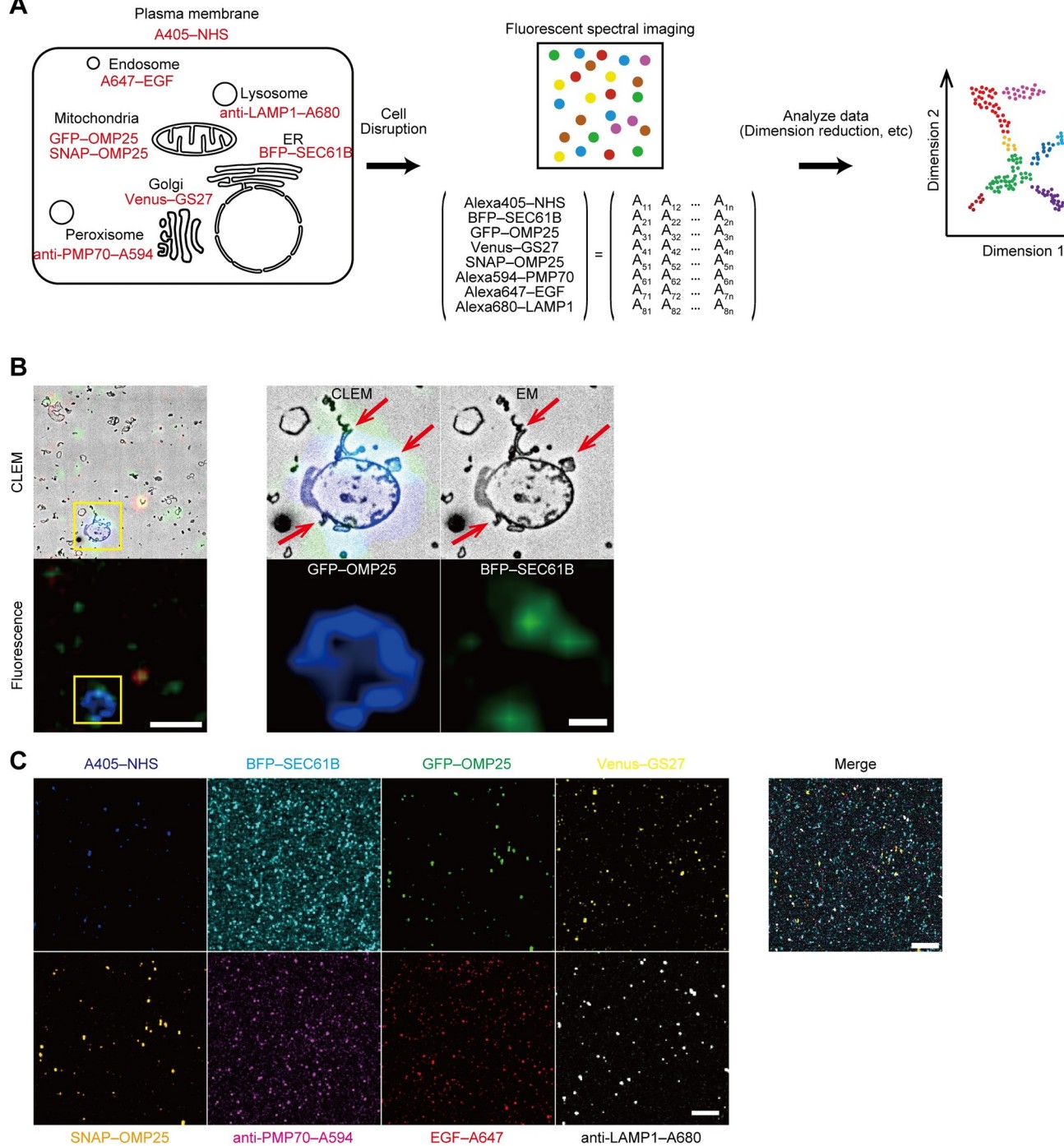

**Fig 1. Multiparametric single-particle analysis.** (**A**) Workflow of multiparametric single-particle analysis of typical organelles. Markers used for labeling organelles are indicated in red. A405, A594, A647, and A680 indicate the fluorescent dyes Alexa Fluor 405, Alexa Fluor 594, Alexa Fluor 647, and Alexa Fluor 680, respectively. Fluorescence intensities of organelle particles for each marker were obtained as eight-dimensional data, as shown in the matrix, which were subjected to dimension reduction for visualization in a two-dimensional map. (**B**) CLEM of organelle particles. Magnified images are shown in the right panels. Blue, GFP–OMP25 (mitochondria); green, BFP–SEC61B (ER); and red, Alexa405–NHS (plasma membrane). Red arrows indicate ER fragments associated with mitochondria. Scale bar, 10 μm and 2 μm (magnified images). (**C**) Unmixing of 8-color fluorescent spectral images and their merged image. Scale bar, 50 μm. CLEM, correlative light and electron microscopy; EGF, epidermal growth factor; ER, endoplasmic reticulum; NHS, *N*-hydroxy-succinimidyl esters.

identify organelle particles in the obtained fluorescence images, Gaussian mixture modeling was applied to the fluorescence intensity of each pixel, thus determining the threshold for separating organelle signals from the background. Using the determined threshold, each image was binarized, and the images of the 8 colors were merged to obtain an image of all particles.

To analyze the fluorescence images, the sum of pixel intensities of each organelle marker was determined for each particle, and eight-dimensional data were thus acquired (Fig 1A and S1 Data). Fluorescent signals of 36,195 particles from 3 independent experiments were combined and dimensionally reduced to be visualized in a two-dimensional plane by principal component analysis (PCA) or uniform manifold approximation and projection (UMAP). While PCA resulted in 1 cluster overall (S2A Fig), UMAP separated the data into 7 clusters (Fig 2A). The numbers of particles classified in each cluster were as follows: Cluster 1, 21,522; Cluster 2, 5,028; Cluster 3, 2,517; Cluster 4, 2,481; Cluster 5, 1,806; Cluster 6, 1,702; and Cluster 7, 1,139. In UMAP, cluster size is dependent on various factors, including the number of data points, their variability, and hyperparameters (a parameter used to tune dimensionality reduction). The particles in each cluster showed different properties of organelle markers: Cluster 1 corresponded to the ER (SEC61B), Cluster 2 to peroxisomes (PMP70), Cluster 3 to mitochondria (GFP–OMP25 and SNAP–OMP25), Cluster 4 to early endosomes (EGF), Cluster 5 to the plasma membranes (Alexa405–NHS), Cluster 6 to the Golgi (GS27), and Cluster 7 to lysosomes (LAMP1) (Fig 2B and 2C and S2 Data). The clusters with high fluorescence intensity of the mitochondrial markers GFP–OMP25 and SNAP–OMP25 coincided with Cluster 3, verifying the specificity of this method. Additionally, as particles from 3 independent experiments were included in all clusters and no cluster unique to specific experiments was found, the classification of these particles was considered reproducible (S2B Fig). These results suggest that organelle particles retain their original membranous components and were accurately detected by this multiparametric particle-based method.

Next, we determined whether exogenous markers truly represented endogenous organelles in our multiparametric particle-based analysis. Organelle particles were prepared from HeLa cells expressing BFP–SEC61B and either GFP–VAMP7 or PEX3–GFP and were labeled with Alexa405–NHS and both anti-PMP70–A594 and anti-LAMP1–A680 antibodies (S3 Fig). Particles were extracted from the captured images, and the five- or six-dimensional data were embedded into two-dimensional planes using UMAP. The analysis revealed clusters based on the marker proteins (S4A and S4B Fig and S3 and S4 Data) (note that although most of these organelle markers were used in the experiment in Fig 2, independent UMAP generated different distribution patterns). In these UMAP spaces, exogenous GFP–VAMP7 and endogenous LAMP1 were found in Cluster 3 (S4C Fig and S3 Data). The distribution of GFP–VAMP7 in Cluster 3 was slightly different from that of LAMP1 in Cluster 3. This may reflect preferential localizations of VAMP7 to late endosomes and of LAMP1 to mature lysosome [28,29]. In addition, exogenous PEX3–GFP colocalized well with endogenous PMP70 in Cluster 2, representing peroxisomes (S4D Fig and S4 Data). These results suggest that even when exogenous organelle markers are overexpressed, UMAP embedding accurately reflects the authentic distribution of these markers.

The applicability of the multiparameter particle-based analysis was further validated in cell lines other than HeLa cells. HEK293T cells expressing mTagBFP2–SEC61B and GFP–OMP25 were loaded with Alexa647–EGF, and their organelle particles were labeled with anti-PMP70–Alexa594 and anti-LAMP1–Alexa680 antibodies (S5 Fig). The analysis of UMAP embedding revealed clusters based on the marker proteins (S6 Fig and S5 Data). Thus, organelle particles in HEK293T cells were correctly detected, and their original identities were preserved during the detection process. These findings suggest that the multiparameter particle-based analysis is applicable across different cell types.

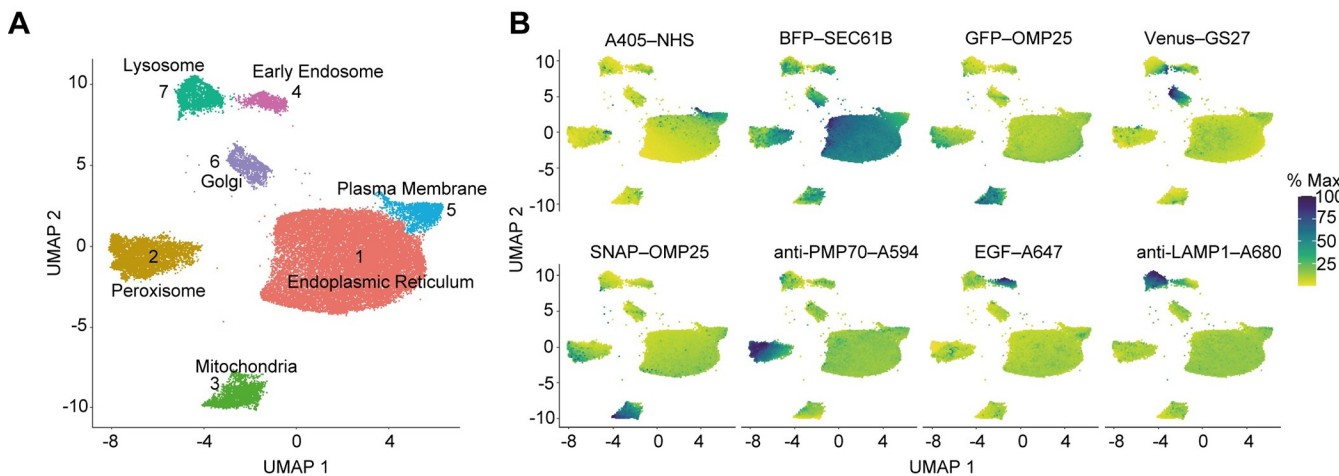

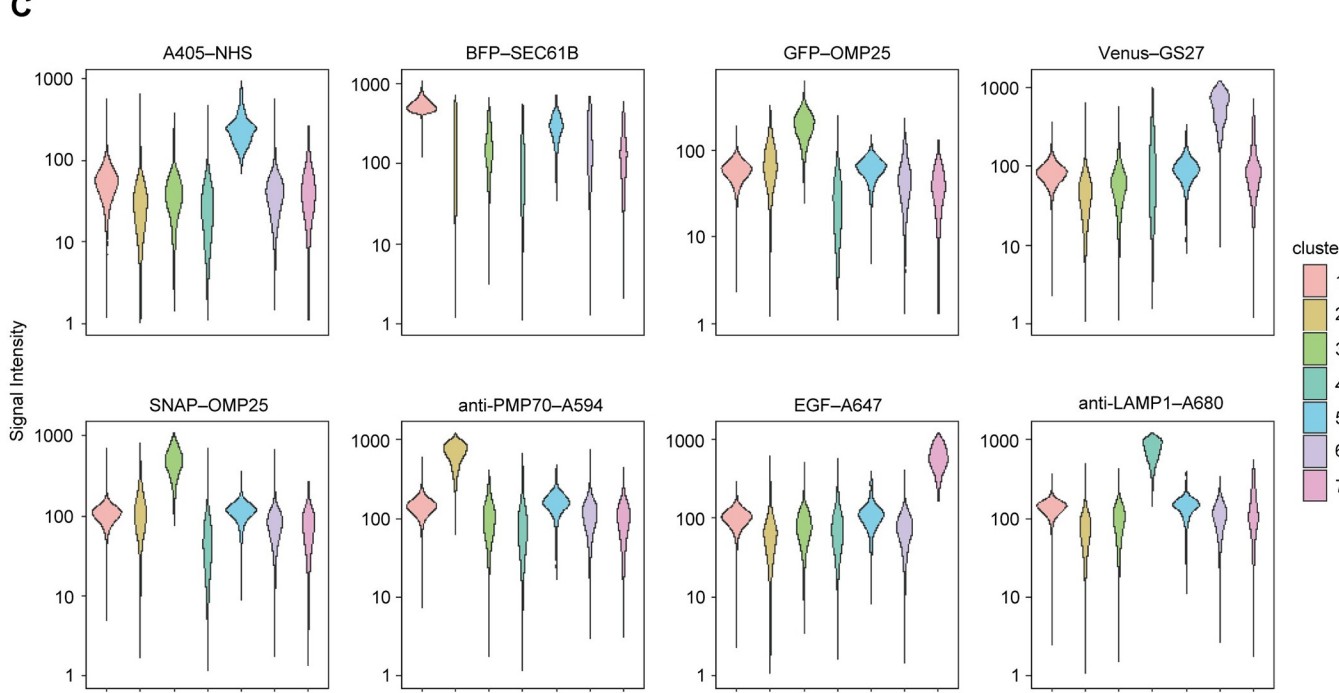

**Fig 2. Multiparametric single-particle analysis of typical organelles.** (**A**) UMAP embedding of the data obtained from 8-color fluorescent images of particles of 7 typical organelles derived from HeLa cells. The numbers of particles classified in each cluster were as follows: Cluster 1, 21,522; Cluster 2, 5,028; Cluster 3, 2,517; Cluster 4, 2,481; Cluster 5, 1,806; Cluster 6, 1,702; and Cluster 7, 1,139. (**B**) The intensities of the fluorescent markers. Particles were colored according to the fluorescence intensity of each marker. The maximum fluorescence intensity in each marker was set to 100%. (**C**) Violin plots of signal intensities (arbitrary unit) of the organelle markers on particles in each cluster. Data obtained from 8-color fluorescent images of particles of 7 typical organelles derived from HeLa cells can be found in S1 Data (Fig 2A) and S2 Data (Fig 2C).

## Applications for the detection of minor organellar populations: ER–mitochondria contact sites

Most organelle markers were enriched in one of the clusters, but the ER marker SEC61B was mixed into several clusters, including mitochondrial clusters (Cluster 3) (Fig 2B). This small population was posited to correspond to the ER–mitochondria contact sites. We therefore

sought to visualize the ER–mitochondria contact sites using a split-GFP-based reporter [30] (Fig 3A). This reporter consists of the first to 10th β-sheets of GFP (GFP1–10) fused to the N terminus of the ER membrane protein ERj1 and the 11th β-sheet of GFP (GFP11) fused to the N terminus of the mitochondrial outer membrane protein TOMM70. When GFP1–10 and GFP11 were associated at the ER–mitochondria contact sites, GFP fluorescence was emitted (S7A Fig). We introduced the fluorescent markers for the ER (BFP–SEC61B) and mitochondria (SNAP–OMP25) into cells expressing this contact site reporter and labeled early endosomes and the plasma membrane with Alexa647–EGF and Alexa405–NHS, respectively. After the preparation of organelle particles, peroxisomes and lysosomes were labeled with Alexa594-conjugated anti-PMP70 and Alexa680-conjugated anti-LAMP1 antibodies, respectively, and 7-color fluorescence images were thus obtained (Figs 3A, S7B, and S7C).

To obtain clusters of the 6 markers (except the split-GFP-based contact site reporter), we used the 6-color dataset obtained in the previous 8-color analysis (BFP–SEC61B, SNAP–OMP25, Alexa647–EGF, Alexa405–NHS, Alexa594–anti-PMP70 antibody, and Alexa680–anti-LAMP1 antibody) as reference data (S6 Data). We embedded these reference data in a new two-dimensional plane using UMAP, resulting in 5 clusters (Fig 3B and S6 Data). Examination of the fluorescence intensity of each marker in the UMAP space revealed that the clusters were formed to contain different organelle markers (S8A Fig and S6 Data). These clusters were detected in all 3 independent experiments, validating the reproducibility of this experiment (S8B Fig). When the query data from cells expressing the ER–mitochondria contact site reporter were annotated using metric learning with the reference data, all 5 of the clusters were mapped with the query data (Fig 3C and S7 Data). Monitoring the fluorescence intensity of each marker in the plotted query data revealed that each cluster primarily contained a distinct marker, as observed among the reference data (S9A Fig and S7 Data). As these clusters were also detected in all 3 independent experiments, the particle classification was considered reproducible (S9B Fig).

We then plotted the GFP signals derived from the ER–mitochondria contact sites onto the UMAP space. Particles with high GFP intensities were detected within both the mitochondrial and ER clusters (Fig 3D). These particles were found in the areas with high BFP–SEC61B signal in the mitochondrial cluster and high SNAP–OMP25 signal in the ER cluster (red arrows in S9A Fig). Thus, those GFP-positive particles were considered to be ER–mitochondria contact sites. Moreover, in our CLEM data on organelle particles, ER fragments were often detected on the mitochondrial surface (Fig 1B). In most cases, ER fragments were much smaller than associated mitochondria particles. Therefore, it is reasonable that the signals of the ER–mitochondria contact sites were detected mainly in the mitochondrial cluster. In summary, the present method can be applied to detect small organelle populations like organelle contact sites in the organelle landscape.

## Applications to organelles in transition: A diagram of endosomes

The successful detection of the ER–mitochondria contact sites suggested that not only typical organelles but also organelles during transition or maturation could be investigated with this method. Accordingly, we analyzed the endocytic pathway, which contains various organelles in transition, after incorporation of EGF (a cargo representative of lysosomal degradation) and transferrin (a cargo representative of recycling to the plasma membrane).

To assign organelles more precisely, we used fluorescent cargos in addition to static organelle markers. GFP–RAB5 (early endosome marker), Venus–RAB11 (recycling endosome marker), and SNAP–RAB7 (late endosome marker) were expressed in HeLa cells. These HeLa cells were incubated with Alexa594-conjugated transferrin (hereafter, Alexa594–transferrin)

A

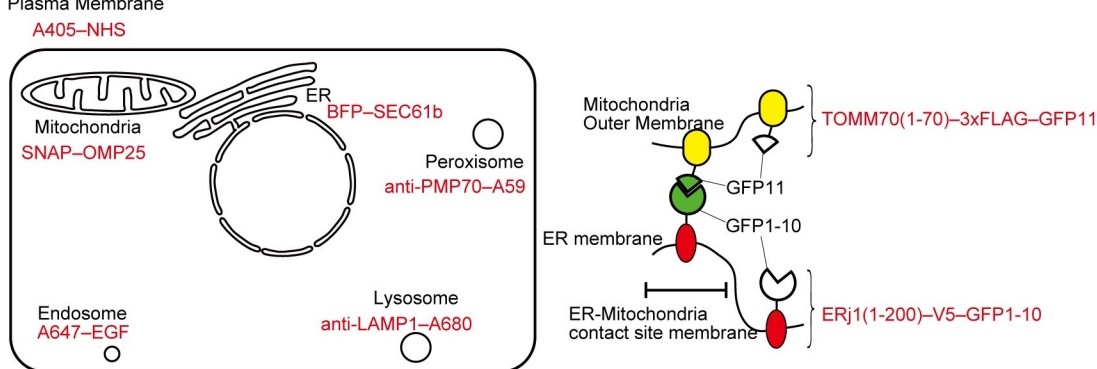

B

C

D

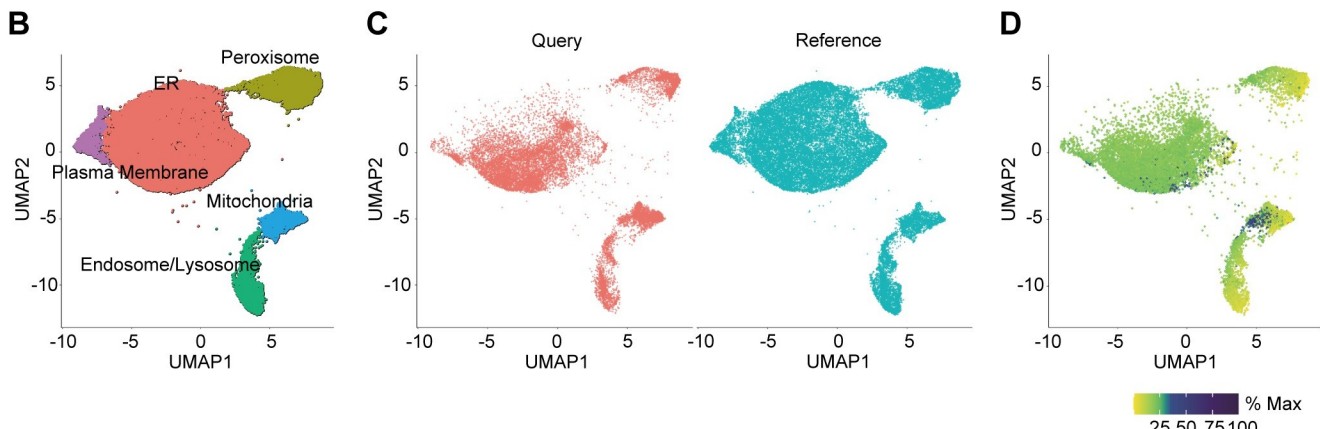

Fig 3. Multiparametric single-particle analysis of the ER–mitochondria contact site. (A) Schematic illustration of cells labeled with the ER–mitochondrial contact site reporter and 6 organelle markers (left) and the split-GFP-based ER–mitochondria contact site reporter (right). ERj1(1–200)–V5–GFP1–10 and TOMM70(1–70)–3×FLAG–GFP11 are only assembled on the ER–mitochondria contact site to form GFP. Thus, those reporters that are not assembled on the ER–mitochondria contact site do not produce GFP signals. (B) UMAP embedding and clustering of the data obtained from fluorescent images of particles labeled with 6 organelle markers as references. The numbers of particles classified in each cluster were as follows: ER, 22,233; peroxisome, 5,088; endosome and lysosome, 3,583; mitochondria, 2,243; the plasma membrane, 1,711. (C) Plot of the data of the experiments with the ER–mitochondrial contact site marker as query using metric learning with the UMAP results in (B) as reference. The numbers of particles plotted on the query were 17,479, and those of references were 34,858. (D) Plot of the GFP signal intensity derived from the ER–mitochondria contact site reporter. The maximum fluorescence intensity of GFP was set to 100%. For visualization, the square root of % max has been plotted. Data obtained from 8-color fluorescent images of particles labeled with 6 organelle markers as references can be found in S6 Data. Data of the experiments with the ER–mitochondrial contact site marker as query can be found in S7 Data.

and Alexa647–EGF at 4°C for 30 min. Then, excess cargos were washed away, while the remainder was internalized by endocytosis at 37°C (Fig 4A). At each time point of endocytosis, we collected organelle particles, stained them with antibodies against LAMP1, and obtained 6-color fluorescence images (S10A–S10D Fig and S8 Data).

To analyze the temporal changes of the features of only EGF- or transferrin-positive particles, we strictly identified these particles as follows (because many endosomes contained neither EGF nor transferrin). We measured the background intensities of particles derived from cells not treated with Alexa594–transferrin and Alexa647–EGF and set the 99th percentile point for signal strength as the threshold for the Alexa594–transferrin and Alexa647–EGF signals (S10E Fig, bottom panels). We then extracted particles that were positive for EGF or

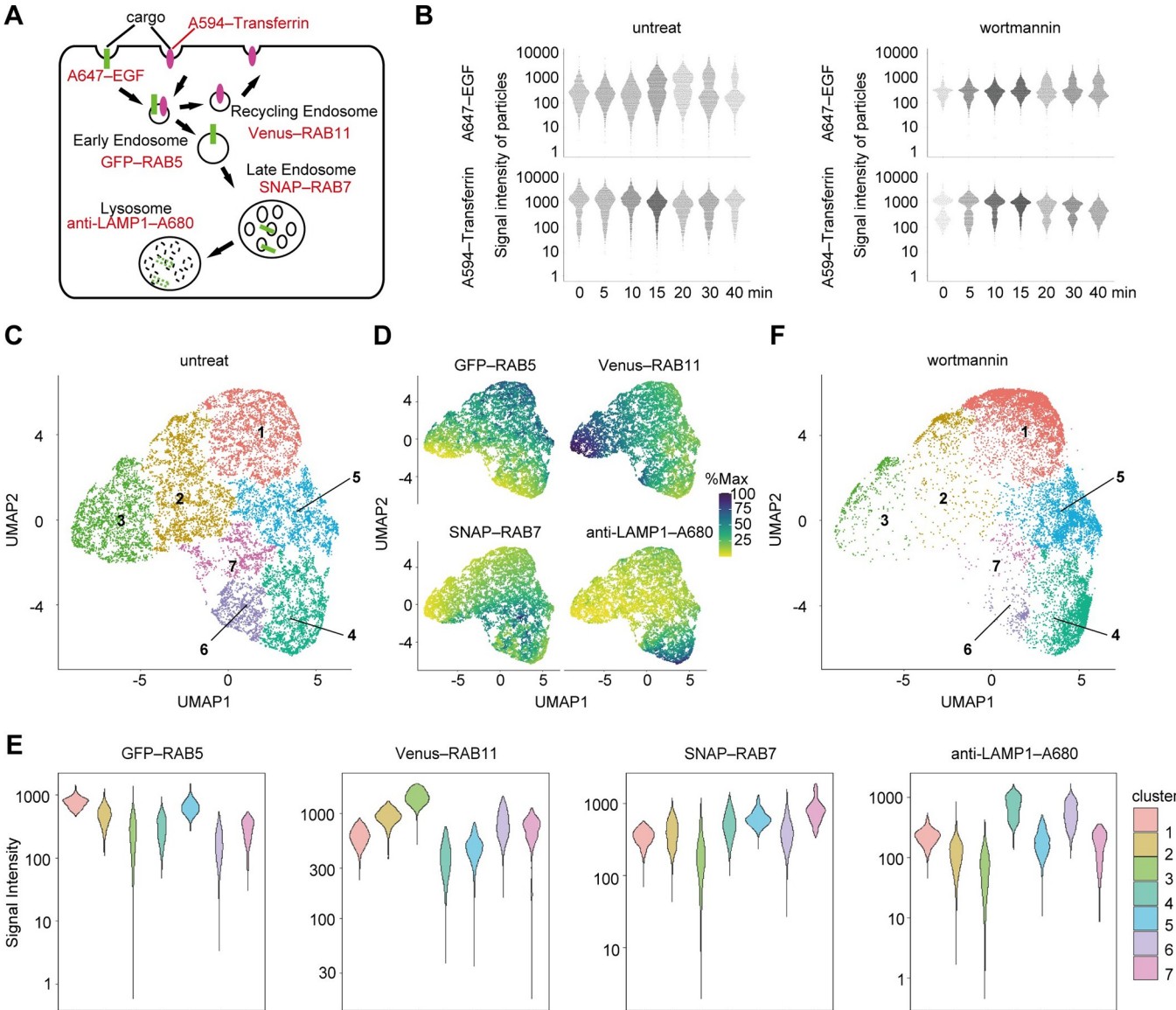

**Fig 4. Multiparametric particle-based analysis of endocytic vesicles containing EGF or transferrin.** (**A**) Schematic illustration of cells used for the analysis of endocytic compartments. Green and magenta indicate A647–EGF and A594–transferrin signals, respectively. Cells were treated with wortmannin or left untreated, followed by treatment with A647–EGF and A594–transferrin for 30 min at 4°C, washing, and incubation at 37°C for 0, 5, 10, 15, 20, 30, and 40 min. (**B**) Signal intensities of Alexa647–EGF and Alexa594–transferrin on particles positive for either EGF or transferrin over the time course (after the shift to 37°C). For wortmannin-untreated cells, the numbers of particles plotted at each time point were as follows: 0 min, 1,820; 5 min, 2,281; 10 min, 2,758; 15 min, 4,394; 20 min, 1,946; 30 min, 2,681; 40 min, 1,046. For wortmannin-treated cells, the particle numbers were as follows: 0 min, 461; 5 min, 2,050; 10 min, 3,415; 15 min, 3,329; 20 min, 1,534; 30 min, 2,369; 40 min, 2,017. (**C**) UMAP embedding of 17,113 data points obtained from fluorescent images of particles of endosomes containing EGF or transferrin. The numbers of particles classified in each cluster were as follows: Cluster 1, 3,347; Cluster 2, 3,092; Cluster 3, 2,959; Cluster 4, 2,869; Cluster 5, 2,423; Cluster 6, 1,374; Cluster 7, 1,049. (**D**) The intensities of the indicated fluorescent markers on the UMAP space. Particles are colored according to the fluorescence intensity of each marker. The maximum fluorescence intensity of each marker was set as 100%. (**E**) Violin plots of the signal intensities of the organelle markers on particles in each cluster. (**F**) A plot of the 15,197 data points of the experiments with the wortmannin treatment as a query using metric learning with the UMAP results in (C) as reference. The numbers of particles classified in each cluster were as follows: Cluster 1, 6,405; Cluster 2, 897; Cluster 3, 623; Cluster 4, 3,637; Cluster 5, 3,145; Cluster 6, 295; Cluster 7, 173. Data obtained from fluorescent images of particles of endosomes containing EGF or transferrin can be found in S8 Data (Fig 4C and 4D) and S9 Data (Fig 4E).

transferrin and tracked the EGF and transferrin signals in this population at each time point. A bright EGF population appeared 15 min after culture conditions were shifted to 37˚C, possibly owing to the fusion of early endosomes, which increased the EGF fluorescence intensity per particle [31] (Fig 4B, left upper panel). The bright population decreased after 30 min, suggesting that Alexa647–EGF was degraded in lysosomes. This observation was consistent with previous reports that EGF reaches lysosomes approximately 30 min after incorporation [32]. To examine the impact of perturbations on EGF endocytosis, we treated cells with wortmannin, an inhibitor of the class III PI3 kinase complex, for 15 min before EGF addition. In wortmannin-treated cells, a decrease in the bright EGF population was not observed, indicating a delay in degradation (Fig 4B, right upper panel). In contrast to EGF, transferrin showed only a slight decrease after 10 min. This probably indicates a low homotopic fusion rate in RAB11-positive recycling endosomes [33] and recycling into the plasma membrane, which is consistent with the reported 10-min half-life of transferrin recycling [34] (Fig 4B, left lower panel).

We then integrated the time-course data of untreated cells from 3 independent experiments, reduced the dimensions of the fluorescence intensity data of 4 endosomal markers (excluding EGF and transferrin) into a two-dimensional UMAP plane, and performed clustering. This resulted in 7 clusters (Fig 4C and S8 Data), all of which were detected in 3 independent experiments, supporting the reproducibility of this particle classification (S11 Fig). These clusters roughly reflected the fluorescence intensity of each endosomal marker (Fig 4D and 4E and S8 and S9 Data). Cluster 1 was considered to comprise early endosomes based on high RAB5 fluorescence intensity, and Clusters 2 and 3 were considered to comprise recycling endosomes based on high RAB11 fluorescence intensity. Cluster 5, with high RAB5 and RAB7 signal intensity, was considered to be a population of endosomes undergoing RAB conversion, during which RAB5 is replaced by RAB7. Clusters 4, 6, and 7, with high signal intensities of RAB7 and/or LAMP1, were considered late endosomes and lysosomes. Next, we embedded the data from the wortmannin-treated cells into the UMAP space by metric learning, using the data from untreated cells as a reference (Fig 4F). The data were separated into 7 clusters, resembling the clustering of the untreated cells. However, the number of particles in Clusters 2, 3, 6, and 7 was reduced, suggesting that the inhibition of the recycling and degradation pathways by wortmannin was adequately represented in the UMAP space.

Next, we extracted the fluorescence intensities of EGF and transferrin from the original data and plotted them with the embedded data at each time point (Fig 5A and 5B). EGF was predominantly present in Clusters 1 and 5 at 0 to 10 min and moved to Clusters 4, 6, and 7 after 15 min. Transferrin was predominantly present in Clusters 1 and 2 at 0 min and migrated to Cluster 3 at 5 to 10 min. After 20 min, there were fewer particles with high transferrin signals. These results suggest that our method successfully visualized the continuous diagram of the endocytic process of EGF and transferrin.

To further characterize each cluster, we analyzed the temporal transition of the 2 cargos. We counted the number of EGF- or transferrin-positive particles in each cluster and plotted the proportion over time (Fig 5C and 5D and S10 Data). Regarding EGF-positive particles, Clusters 6 and 7 showed the highest proportions at 15 min (Fig 5C). These clusters were considered to be late endosomes positive for RAB7, as they did not show high RAB5 signal intensity (Fig 4D and 4E). The proportion of Cluster 4 increased over time, indicating that it represents lysosomes (Fig 5C). In Cluster 4, LAMP1 was more enriched than RAB7 (Fig 4D and 4E). The proportion of transferrin-positive particles in Clusters 6 and 7 increased at 10 min, and that of Cluster 4 increased at 15 min (Fig 5D). These data are consistent with Clusters 6 and 7 representing late endosomes and Cluster 4 representing lysosomes because some population of transferrin receptors are directed to lysosomal degradation [35–37].

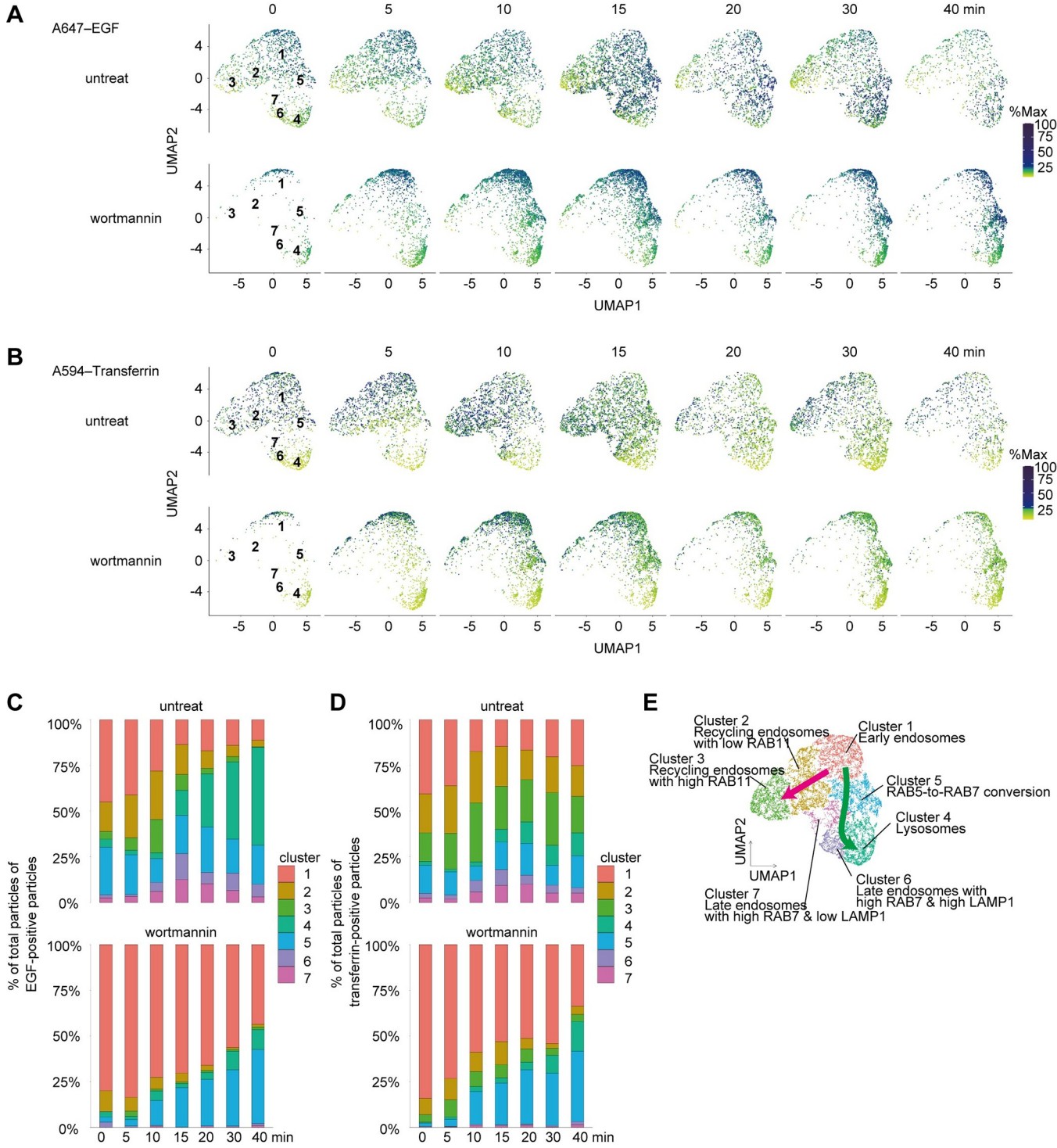

**Fig 5. Changes in the distribution of clusters during the endocytic process.** (**A**, **B**) Distribution of Alexa647–EGF and A594–transferrin intensities of particles along the time course with or without wortmannin. The colors indicate the signal intensity of EGF or transferrin. The maximum fluorescence intensity of each marker was set to 100%. For visualization, the square root of %max has been plotted. (**C**, **D**) Stacked bar graphs showing the proportion of clusters in EGF-positive particles (**A**) and transferrin-positive particles (**B**) at each time point (after the shift to 37°C) from wortmannin-treated (lower panels) and untreated (upper panels) cells. (**E**) Characterization of the clusters and putative pathways depicted in UMAP space. The magenta arrow indicates the putative recycling pathway, and the green arrow indicates the degradation pathway. Data obtained from fluorescent images of particles of endosomes containing EGF or transferrin and extracted EGF- or transferrin-positive particles can be found in S10 Data.

Upon treatment with wortmannin, EGF was predominantly distributed in Clusters 1 and 5, while EGF in Clusters 6 and 7 was reduced, indicating inhibition of transport to late endosomes and lysosomes (Fig 5C). Transferrin was also predominantly distributed in Clusters 1 and 5, with a decrease in Clusters 2 and 3 (Fig 5D). This is consistent with previous research showing that transferrin localizes to RAB5-positive structures instead of RAB11-positive structures upon PI3 kinase inhibition [38–41]. By tracking the intensities of EGF and transferrin, we were able to classify endocytic organelles in detail, which could not be determined simply by detecting organellar markers (Fig 5E).

## Discussion

We developed a multiparametric particle-based method that simultaneously detects multiple organelles with sufficient resolution. This method also minimizes observer biases because it analyzes organelle particles from a large number of cells. With this method, we can monitor the landscape of organelles in UMAP space.

This multiparametric particle-based analysis is effective when a large number of (typically more than 4 or 5) markers are observed simultaneously. In addition, normalization of signal intensity across different datasets is straightforward, enabling the combined analysis of multiple datasets, as in our analysis of ER–mitochondria contact sites and endocytosis. We were able to exploit this advantage to extract and define intermediates of transitional organelles such as endosomes (Fig 5E). While population analysis for EEA1, APPL1, RAB5, and RAB7 combined with CLEM has been reported, only 2 endosome markers were used for each electron microscopy image [42]. Therefore, experiments must be performed separately for each of these marker pairs. For late endosomes and lysosomes, counting structures positive for RAB7 and LAMP1 after dextran incorporation has been reported [43]. In contrast, our method using 4 endosomal markers and 2 representative cargos (for recycling and degradation, respectively) can monitor endocytosis processes from early endosomes to recycling endosomes, late endosomes, and lysosomes all at once. A further potential application of our method would be to measure how the levels of key molecules in an organelle change during its differentiation or maturation. For example, the levels of PI4P and syntaxin 17 change during autophagosome maturation [44], which can be better demonstrated by the present method using multiple markers for each stage of autophagosome formation and maturation, as autophagosomes at different stages coexist in cells. In such cases, our single-particle analysis method, which examines the state of individual autophagosomes, would be more appropriate than biochemical methods that examine averages. In addition, it is difficult to quantitatively analyze many organelle structures in cells using fluorescence microscopy. Our particle-based analysis method can overcome this problem.

This cell-free technique is applicable not only to fluorescence microscopy but also to flow cytometry. By combining flow organelle sorting and highly sensitive proteomic or lipidomic analysis, it would be possible to identify new organelle entities. However, the detection limit of particle size by flow cytometry is approximately 100 nm in diameter, hindering the detection of small organelles [45]. In addition, multiple organelles can be included in a single droplet during sorting, making purification difficult. If high-throughput, multiparametric microparticle sorting becomes possible with microfluidics [46], it may be applicable to organelle isolation.

One clear limitation of this technique is that the original morphology and spatial information are lost. To compensate for this shortcoming, combined applications with fluorescence microscopy or CLEM of intact cells are necessary. Additionally, to improve the resolution within a single organelle, for example, to separate distinct organellar subdomains, we can

prolong the sonication time to make the particles smaller. However, because this would reduce the signal-to-background ratio and might destroy organelle contacts, we used sonication conditions that were as mild as possible. To investigate organellar subdomains and fragile contacts, we need to carefully optimize the sonication conditions. Furthermore, the fact that only a small number of antibodies are effective for staining intact organelle particles limits the number of parameters. Current fluorescent particle detection methods utilize spectral multiplexing, but this approach has only been able to detect up to 8 colors. Methods with multiplexed imaging, such as CODEX or DNA-PAINT, could significantly increase the number of targets [6,47]. Our method, which is independent of fixation, is advantageous for the optimization of staining conditions. If organelle particles can be detected using only antibodies without exogenous expression of organelle markers, this method would become particularly applicable to physiological and pathological studies. Just as flow cytometry revolutionized immunology and hematology, multiparametric organelle particle analysis would provide more detailed information about organelles, thereby deepening our understanding of cell biology.

## Materials and methods

### Cell lines and culture conditions

Authenticated HeLa cells and human embryonic kidney (HEK) 293T cells obtained from RIKEN were utilized. The cells were cultured in a 5% $CO_2$ incubator at 37°C, using Dulbecco's Modified Eagle Medium (DMEM; Sigma-Aldrich, D6546) supplemented with 10% fetal bovine serum (FBS; Sigma-Aldrich, 173012) and 2 mM L-glutamine (Gibco, 25030–081).

### Plasmids

The respective cDNAs encoding EGFP (Clontech, 6084–1), SNAP-tag (New England BioLabs, N9181S), HaloTag (Promega, N2701), OMP25 [25] (the signal sequence for mitochondria outer membrane targeting, obtained by PCR with the primers encoding the signal sequence), human SEC61B (NM_006808, obtained from 293T cells), human GS27 (AF007548, obtained from HeLa cells), human RAB5 (AF498936, obtained from HeLa cells), mouse RAB7 (AB232599, gifted from Prof. M. Fukuda), human RAB11 (NM_004663, obtained from HeLa cells), TOM70(1–70)–3×FLAG–GFP11, ERj1(1–200)–V5–GFP1–10 [30] (gifted from Prof. Y. Tamura), human PEX3 (NM_003630), and mouse VAMP7 (NM_011515.4) were subcloned into pMRX-IP vector [48,49].

### Antibodies and reagents

Rabbit polyclonal anti-LAMP1 (Abcam, ab24170) and anti-PMP70 (Thermo Fisher Scientific, PA1-650) antibodies were labeled with Alexa Fluor 680 and 594 NHS ester (Thermo Fisher Scientific, A20008, A20004), respectively, and then free fluorescence molecules were removed using Pierce Dye removal Columns (Thermo Fisher Scientific, 22858) according to the manufacturer's instructions. Alexa Fluor 405 NHS ester (A3000), Alexa Fluor 647 EGF complex (E35351), and Alexa Fluor594-conjugate human transferrin (T13343) were obtained from Thermo Fisher Scientific. SNAP-Cell TMR-star (New England BioLabs, S9105S) and HaloTag ligand CF650 (GoryoChemical A308-02) were used to label SNAP-tag and HaloTag, respectively. Wortmannin (W1628) was purchased from Sigma-Aldrich.

### Stable expression in HeLa cells by retrovirus-mediated transfection

To prepare the retrovirus solution, HEK293T cells were transfected with the pMRX-IP-based retroviral plasmid, along with pCG-gag–pol and pCG-VSV-G (a gift from Prof. T. Yasui),

using Lipofectamine 2000 (Thermo Fisher Scientific, 11668019) for a period of 4 to 6 h. Following transfection, the cells were cultured in DMEM for 2 to 3 d. The medium containing the retrovirus was collected and filtered through a 0.45-μm filter unit (Ultrafree-MC; Millipore). The filtered medium was then added to HeLa or HEK293T cells for infection, along with 8 μg/ml polybrene (Sigma-Aldrich, H9268). Viral infections were conducted to obtain cells expressing single or multiple organelle markers. Multiple-organelle marker coexpressing cells were isolated using a cell sorter (MoFlo Astrios EQ, Beckman Coulter).

## Preparation of organelle particles

Eight-color typical organelle particles were prepared as follows. HeLa cells coexpressing mTag2BFP2–SEC61B, EGFP–OMP25, Venus–GS27, and SNAP-tag–OMP25 were subconfluently cultured in a 6-cm dish. The cells were washed 3 times with serum-free DMEM and then incubated at 37°C for 5 min with 400 ng/ml Alexa647–EGF. Subsequently, the cells were immediately washed 3 times with serum-containing DMEM, followed by 3 washes with ice-cold PBS. Afterward, the cell membrane was labeled by incubating the cells at 4°C for 15 min with 8 μg/ml Alexa405–NHS. The cells were then washed twice with ice-cold PBS and with HEPES buffer (20 mM HEPES-KOH (pH 7.4), 250 mM sucrose, 1 mM EDTA) and were then collected by adding 1 ml of HEPES buffer with protease inhibitor (Nacalai Tesque, 03969–34). The cells were sonicated 3 times for 1 s each using a sonicator (TAITEC, VP-050N). The supernatant was obtained by centrifugation at $1,500 \times g$ for 10 min, repeated twice. To this supernatant, 0.3% BSA, 125 nM SNAP-Cell TMR-star, Alexa594-conjugated anti-PMP70, and Alexa680-conjugated anti-LAMP1 antibodies were added and incubated at 4°C for 1 h. The multicolor organelle suspension was mounted on poly-L-lysine-coated coverslips (Matsunami, C1110) with ProLong Gold Antifade Mountant (Thermo Fisher Scientific, P36934). For the experiment investigating mitochondria–ER contact sites, HeLa cells coexpressing mTagBFP1–SEC61B, TOM70(1–70)–3×FLAG–GFP11, ERj1(1–200)–V5–GFP1-10, and SNAP-tag–OMP25 were prepared for a 6-color organelle suspension according to the methods described above. For the experiment examining the process of endocytosis, HeLa cells coexpressing EGFP–RAB5, Venus–RAB11, and SNAP-tag–RAB7 were cultured overnight in 7 dishes with serum-free DMEM. After incubation with or without 200 nM wortmannin at 37°C for 15 min, 1 μg/ml Alexa647–EGF and 5 μg/ml Alexa594–transferrin were added and incubated at 4°C for 30 min. After 2 washes with ice-cold PBS and with serum-free DMEM, the cells were incubated at 37°C for 0, 5, 10, 20, 30, and 40 min with or without wortmannin. The cells in the 7 dishes were washed twice with ice-cold PBS and twice with HEPES buffer. Cell lysis was obtained as described above, followed by incubation of the lysates at 4°C for 1 h with 125 nM SNAP-Cell TMR-star and Alexa680-conjugated anti-LAMP1 antibody. The 6-color labeled organelle suspension was mounted on coverslips using the same procedure. Sample preparations under these respective conditions were repeated 3 times. As reference samples for spectral imaging and linear unmixing, each single-color organelle sample was prepared using the same methods, either with cells expressing single-organelle markers or with cells not expressing any exogenous organelle markers.

## Fluorescence spectral imaging and linear unmixing

Fluorescence spectral images were acquired using an Olympus FV3000 confocal laser microscope equipped with a 4-channel cooled GaAsP photomultiplier and a diffraction grating. The microscope was equipped with a 60× oil-immersion objective lens (NA 1.42, UPLAXAPO; Olympus). Fluorescent images were captured using FLUOVIEW software (FV31-SW, ver. 2.4.1.198, Olympus). All fluorophores were excited sequentially using 640 nm, 488 nm, and

561 nm lasers as well as a 405-nm laser with a 405/488/561/640 nm multiband beam splitter. Fluorescent signals were separated with dichroic mirrors (SDM400–540, SDM400–470, SDM400–620) and were collected onto 4 GaAsP detectors with gating in lambda mode every 5 nm with a 10-nm bandwidth [13 steps from 411 to 481 nm (spectrometer 1), 11 steps from 494 to 554 nm (spectrometer 2), 12 steps from 569 to 634 nm (spectrometer 3), and 16 steps from 646 to 731 nm (spectrometer 4), as shown in S1B Fig]. In order to avoid signal crosstalk between adjacent fluorescence channels, 8 fluorophores with distinct spectral distances were selected, and the samples were irradiated sequentially with lasers in descending order of their wavelengths; that is, fluorescence from 646 to 731 nm was excited by a 640-nm laser, fluorescence from 569 to 634 nm was excited by a 561-nm laser, fluorescence from 494 to 554 nm was excited by a 488-nm laser, and fluorescence from 411 to 481 nm was excited by a 405-nm laser, as shown in S1B Fig. We obtained images of organelle particles labeled with a single fluorophore (each color used for labeling multicolored organelles) using the same image acquisition settings as for multicolored organelles (including laser power and detector settings). The images captured across a total of 52 steps were integrated using the "Append Images" function in FLUOVIEW to create a single lambda series. The multicolor images were subjected to linear unmixing using the "Normal Unmixing" function in FLUOVIEW. Reference spectra generated from the images of single-color organelle using the "Spectral Image Unmixing" function in FLUOVIEW were employed.

## CLEM

For CLEM of fluorescent-labeled organelle particles, the organelle suspension was mounted on a glass bottom dish with 150-μm grids (Iwaki, TCI-3922-035R-1CS). A glass coverslip was carbon-coated using a vacuum evaporator (JEOL, IB-29510VET) and treated with poly-L-lysine for 15 min. The organelle particles were fixed in 2% paraformaldehyde (Nacalai Tesque, 26126–54) and 0.5% glutaraldehyde (TAAB, G018/1) in 0.1 M phosphate buffer at pH 7.4 for 1 h at 4˚C. Fluorescence spectral imaging and linear unmixing were performed as described above. Following fluorescence image acquisition, the organelles were embedded in resin and subjected to trimming, slicing of 25-nm thick serial sections, electron microscopy imaging, and image processing, following previously established protocols [50].

## Particle extraction

Data processing and analysis were performed mostly using ImageJ (v1.54f) in Fiji [51] and R (v4.2.1) in RStudio (v2023.03.1). After linear unmixing, the images were exported in 16-bit grayscale TIFF format using FLUOVIEW (FV31S-SW, Evident). The TIFF images were imported using the EBImage (v4.38.0) package and compiled for each replicate, and the intensity of all pixels was recorded for each channel. For each channel, pixel intensities were classified using a Gaussian Mixture Model as implemented in the R package mclust (v6.0.0). The number of clusters was selected by calculating the Bayesian information criterion (BIC) and selecting the smallest value (in practice, the largest value was selected if the BIC calculation using the GMM function in mclust yielded negative values). Pixel intensities were assigned to clusters by taking the cluster with the highest responsibility for each data point. By assigning the signal intensity of each pixel to clusters, intensities at which the cluster switches were set as crossing points. In all cases, the threshold between particles and the background was determined as the highest crossing point. After determining the thresholds for all channels, particles were extracted using Jython in Fiji (ImageJ). A binary image was created for each channel using the above thresholds, and all 8 channels were merged to create a binary image of the

particles. The fluorescence intensity of the particles was determined using the pixel intensity from the prebinary image, using the total sum of pixel intensities contained within the particles.

When extracting particles from endocytosis experiments, RAB11 signals were excluded from the creation of a binary image of the particles, because the RAB11-positive signal foci were so numerous that the clusters were divided into only 2 clusters based on the presence or absence of RAB11, making it difficult to analyze endocytic particles. For the particle analysis of HeLa cells expressing GFP–VAMP7, SNAP-tag–OMP25 signals were excluded from the creation of a binary image of the particles and data processing of multidimensional data because of unsuccessful spectral unmixing.

### Data analysis

Each experimental dataset was independently tested 3 times, and all results were used together for data analysis. All data were converted to cell_data_set objects using the monocle3 [52] (v1.2.9) package, followed by dimension reduction using PCA, batch processing using the align_cds function, and embedding on a two-dimensional plane using UMAP. Clustering was tested using $k = 60$.

Data analysis of particle data obtained from cells labeled with split-GFP for analysis of ER–mitochondria contact sites and embedded by metric learning was performed as follows. First, six-dimensional data obtained from the previous 8-color analysis, excluding GS27 and GFP–OMP25 signals, were embedded in a two-dimensional plane using UMAP, from which reference data were obtained. These data were then used to map six-dimensional query data obtained from cells labeled with split-GFP for ER–mitochondria contact sites, excluding split-GFP contact site markers, onto UMAP using metric learning. The brightness intensity of GFP was plotted on two-dimensional data embedded in UMAP after converting the seven-dimensional data back to the cell_data_set format, dimension reduction with PCA, batch processing with the align_cds function, and normalization with size factors. Clustering was tested using $k = 20$, thus obtaining 9 clusters. BFP-SEC61B-positive clusters were grouped as 1 cluster manually, and 5 clusters were thus obtained. For visualization, the square root values of GFP signals were plotted.

For particle analysis of endocytosis, the top 1 percentile of the background brightness intensity of Alexa647 and Alexa594 channels extracted from cells without addition of Alexa647-EGF and Alexa594-transferrin was used as a threshold to extract particles from each time point poststimulation. In addition, all points with a value of 0 for either RAB5, RAB7, or RAB11 were removed. All of the above data were processed in the cell_data_set format, reduced in dimension using PCA, batch processed using the align_cds function, and further reduced using UMAP. Clustering was tested using $k = 20$. Particles with mean fluorescence intensity values above the aforementioned thresholds, which were used for determining particle extraction, were used for counting EGF- or transferrin-positive particles. For visualization of EGF or transferrin transition in endocytic pathways, the square root values of Alexa647-EGF or Alexa594-transferrin signals were plotted.

### Use of large language model (LLM) and other AI tools

We utilized ChatGPT, DeepL, and DeepL Write to translate the text from Japanese into English and improve the English text.

### Supporting information

**S1 Fig. Spectral imaging of fluorescently labeled organelle particles; related to Fig 1.** (A) The emission spectra of the fluorophores (Alexa Fluor 405, mTagBFP2, EGFP, Venus,

TMR-Star, and Alexa Fluor 594/647/680) used in confocal microscopy. (**B**) Schematic representation of the equipment employed for 8-color confocal microscopy. Specimens were subjected to excitation by 4 lasers reflected with a dichroic mirror. Fluorescence images were then captured by 4 sets of spectrometers equipped with diffraction gratings and detectors. (**C**) Montage of fluorescence images obtained by spectral imaging of fluorescently labeled organelle particles. Images acquired by shifting the median wavelength (10-nm width) by 5 nm are aligned from the upper left (411 nm) to the lower right (731 nm). Images acquired by spectrometers 1, 2, 3, and 4 are outlined in purple, cyan, green, and red, respectively. Numbers indicate the mean wavelength of each window. Scale bar, 100 μm.
(PDF)

**S2 Fig. Dimension reduction of the 8-color data; related to Fig 2.** (**A**) PCA plots of the data obtained from 8-color fluorescent images of organelle particles derived from HeLa cells. Particles were colored according to the fluorescence intensity of each marker. The maximum fluorescence intensity of each marker was set to 100%. (**B**) The data from 3 independent experiments were plotted on UMAP spaces. The numbers of particles plotted in each experiment were as follows: Experiment 1, 10,749; Experiment 2, 14,286; and Experiment 3, 11,160. Data obtained from 8-color fluorescent images of particles of 7 typical organelles derived from HeLa cells can be found in S1 Data.
(PDF)

**S3 Fig. Spectral imaging and linear unmixing of the images of organelle particles labeled with GFP–VAMP7 or GFP–PEX3.** (**A**, **B**) Montage of fluorescence images obtained by spectral imaging of fluorescently labeled organelle particles. Organelle particles from HeLa cells expressing BFP–SEC61B and GFP–VAMP7 (A) or PEX3–GFP (B) were labeled with Alexa405–NHS and both anti-PMP70–A594 and anti-LAMP1–A680 antibodies and are shown as in S1C Fig. Scale bar, 100 μm. (**C**, **D**) Unmixing results of the fluorescent spectral images in A and B, respectively. Scale bar, 50 μm.
(PDF)

**S4 Fig. Multiparametric single-particle analysis of typical organelles containing GFP–VAMP7 or PEX3–GFP.** (**A**, **B**) UMAP embedding of the data obtained from 6-color fluorescent images of particles of 5 typical organelles derived from HeLa cells described in S3 Fig (expressing GFP–VAMP7 (**A**) or PEX3–GFP (**B**)). The numbers of particles classified in each cluster in A were as follows: Cluster 1, 3,711; Cluster 2, 3,063; Cluster 3, 2,724; Cluster 4, 1,044; Cluster 5, 786; Cluster 6, 317. The numbers of particles classified in each cluster in B were as follows: Cluster 1, 33,167; Cluster 2, 3,553; Cluster 3, 1,926. (**C**, **D**) The intensities of the fluorescent markers. Particles were colored according to the fluorescence intensity of each marker. The maximum fluorescence intensity in each marker was set to 100%. Data obtained from 6-color fluorescent images of particles of 5 typical organelles derived from HeLa cells stably expressing GFP–VAMP7 and PEX3–GFP can be found in S3 and S4 Data, respectively.
(PDF)

**S5 Fig. Spectral imaging and linear unmixing of the images of organelle particles from HEK293T cells.** (**A**) Montage of fluorescence images obtained by spectral imaging of fluorescently labeled organelle particles. HEK293T cells expressing mTagBFP2–SEC61B and GFP–OMP25 were loaded with A647–EGF, and their organelle particles were labeled with anti-PMP70–A594 and anti-LAMP1–A680 antibodies, as shown in S1C Fig. Scale bar, 100 μm. (**B**) Unmixing results of the fluorescence spectral images in A. Scale bar, 50 μm.
(PDF)

**S6 Fig. Multiparametric single-particle analysis of typical organelle particles from HEK293T cells.** (A) UMAP embedding of the data obtained from 6-color fluorescent images of particles of 5 typical organelles derived from HEK293T cells. The numbers of particles classified in each cluster were as follows: Cluster 1, 3,690; Cluster 2, 1,450; Cluster 3, 1,416; Cluster 4, 642; Cluster 5, 577. (**B**) The intensities of the fluorescent markers. Particles are colored according to the fluorescence intensity of each marker. The maximum fluorescence intensity in each marker was set to 100%. Data obtained from 5-color fluorescent images of particles of 4 typical organelles derived from HEK293T cells can be found in S5 Data.
(PDF)

**S7 Fig. Spectral imaging and linear unmixing of images of organelle particles containing an ER–mitochondria contact site marker; related to Fig 3.** (A) Fluorescent images of HeLa cells expressing BFP–SEC61B, GFP–ER–mito [ER–mitochondria contact site marker, ERj1(1–200)–V5–GFP1–10 and TOMM70(1–70)–3×FLAG–GFP11], and SNAP–OMP25. Scale bars, 10 μm and 2 μm (inset). (**B**) Montage of fluorescence images obtained by spectral imaging of fluorescently labeled organelle particles. Images were acquired and are shown as in Extended Data Fig 1C. (**C**) Unmixing results of fluorescent spectral images in B. Scale bar, 100 μm.
(PDF)

**S8 Fig. Distribution of organelle markers of the reference data in UMAP space; related to Fig 3.** (A) Intensities of fluorescent markers from the reference data shown in Fig 3. Particles were colored according to the fluorescence intensity of each marker. The maximum fluorescence intensity in each marker was set to 100%. (**B**) UMAP embedding of the data of the reference obtained from 3 independent experiments. The numbers of particles plotted on each experiment were as follows: Experiment 1, 10,387; Experiment 2, 13,830; and Experiment 3, 10,641. Data obtained from 8-color fluorescent images of particles labeled with 6 organelle markers as references can be found in S6 Data.
(PDF)

**S9 Fig. Distribution of organelle markers of the query data of Fig 3 in UMAP space; related to Fig 3.** (A) Intensities of the fluorescent markers from the query data. Red arrows indicate split-GFP-positive particles. (**B**) UMAP embedding of the data of the query obtained from 3 independent experiments. The numbers of particles plotted in each experiment were as follows: Experiment 1, 7,245; Experiment 2, 4,160; and Experiment 3, 6,065. Data of the experiments with the ER–mitochondrial contact site marker as query can be found in S7 Data.
(PDF)

**S10 Fig. Spectral imaging and linear unmixing of the images of organelle particles labeled with endocytosis-related markers; related to Fig 4.** (A, **B**) Montage of fluorescence images obtained by spectral imaging of fluorescently labeled endocytic particles without (A) and with (B) wortmannin treatment. Images were acquired and are shown as in S1C Fig. (**C, D**) Unmixing results of the fluorescent spectral images in A and B. (**E**) Histogram of the signal intensity of particles from cells treated or untreated with A647–EGF and A594–transferrin. The 99th percentile point for untreated samples is indicated by the red dashed line.
(PDF)

**S11 Fig. Reproducibility of the data and marker intensities in each cluster in the endocytosis analysis; related to Fig 4.** (A) Particles from each replicate separately plotted on the UMAP data shown in Fig 4C. The numbers of particles plotted on each experiment were as follows: Experiment 1, 2,893; Experiment 2, 5,791; and Experiment 3, 8,429. (**B**) Distribution of each replicate in Clusters 1–7. Data obtained from fluorescent images of particles of

endosomes containing EGF or transferrin can be found in S8 Data.
(PDF)

**S1 Data. Data for graphs in Figs 2A, 2B, and S2.** Data obtained from 8-color fluorescent images of particles of 7 typical organelles derived from HeLa cells were dimensionally reduced followed by clustering. Results for experimental replicates, fluorescent intensities of organelle markers on particles, PCA1, PCA2, UMAP1, UMAP2, clusters, and areas of particles.
(CSV)

**S2 Data. Data for graphs in Fig 2C.** Data obtained from 8-color fluorescent images of particles of 7 typical organelles derived from HeLa cells were dimensionally reduced followed by clustering. Results for experimental replicates, fluorescent intensities of organelle markers on particles normalized by size factors, UMAP1, UMAP2, clusters, and areas of particles.
(CSV)

**S3 Data. Data for graphs in S4A and S4C Fig.** Data obtained from 6-color fluorescent images of particles of 5 typical organelles derived from HeLa cells stably expressing GFP–VAMP7 were dimensionally reduced followed by clustering. Results for experimental replicates, fluorescent intensities of organelle markers on particles areas of particles, clusters, UMAP1, and UMAP2.
(CSV)

**S4 Data. Data for graphs in S4B and S4D Fig.** Data obtained from 5-color fluorescent images of particles of 4 typical organelles derived from HeLa cells stably expressing PEX3–GFP were dimensionally reduced followed by clustering. Results for experimental replicates, fluorescent intensities of organelle markers on particles areas of particles, clusters, UMAP1, and UMAP2.
(CSV)

**S5 Data. Data for graphs in S6 Fig.** Data obtained from 5-color fluorescent images of particles of 4 typical organelles derived from HEK293T cells were dimensionally reduced followed by clustering. Results for experimental replicates, fluorescent intensities of organelle markers on particles areas of particles, clusters, UMAP1, and UMAP2.
(CSV)

**S6 Data. Data for graph in Figs 3B and S8.** Data obtained from 8-color fluorescent images of particles labeled with 6 organelle markers as references were dimensionally reduced followed by clustering. Results for experimental replicates, fluorescent intensities of organelle markers on particles areas of particles, clusters, UMAP1, and UMAP2.
(CSV)

**S7 Data. Data for graphs in Figs 3C (query), 3D, and S9.** Data of the experiments with the ER–mitochondrial contact site marker as query using metric learning with the UMAP results in Fig 3b as reference. Results for experimental replicates, fluorescent intensities of organelle markers on particles areas of particles, UMAP1, and UMAP2.
(CSV)

**S8 Data. Data for graphs in Figs 4C, 4D, 4F, and S11.** Data obtained from fluorescent images of particles of endosomes containing EGF or transferrin. Results for experimental condition, ligand-treated time, fluorescent intensities of organelle markers on particles, areas of particles, experimental replicates, clusters, UMAP1, and UMAP2.
(CSV)

**S9 Data. Data for graphs in Fig 4E.** Data obtained from fluorescent images of particles of endosomes containing EGF or transferrin. Results for experimental replicates, experimental condition, fluorescent intensities of organelle markers on particles normalized by size factors, UMAP1, UMAP2, clusters, and areas of particles.
(CSV)

**S10 Data. Data for graphs in Fig 5C and 5D.** Data obtained from fluorescent images of particles of endosomes containing EGF or transferrin. EGF- or TF-positive particles were extracted and results for experimental condition, fluorescent intensities of organelle markers on particles, areas of particles, experimental replicates, clusters, UMAP1, and UMAP2.
(CSV)

## Acknowledgments

We thank Keiko Igarashi for her technical assistance with the establishment of cell lines expressing organelle markers and analysis of organelle particles, Chieko Saito and Yoko Ishida for their technical assistance with the 3D-CLEM experiments, Shoi Shi for technical consulting with data analysis, Mitsunori Fukuda for providing Rab7 cDNA, Yasushi Tamura for providing cDNAs of mitochondria–ER contact site marker, and Teruhito Yasui for providing pCG-gag-pol and pCG-VSV-G vectors.

## Author Contributions

**Conceptualization:** Yoshitaka Kurikawa, Ikuko Koyama-Honda, Noboru Mizushima.

**Data curation:** Yoshitaka Kurikawa, Ikuko Koyama-Honda.

**Formal analysis:** Yoshitaka Kurikawa, Ikuko Koyama-Honda.

**Funding acquisition:** Ikuko Koyama-Honda, Noboru Mizushima.

**Investigation:** Yoshitaka Kurikawa, Ikuko Koyama-Honda, Norito Tamura, Seiichi Koike.

**Methodology:** Yoshitaka Kurikawa, Ikuko Koyama-Honda, Norito Tamura, Seiichi Koike, Noboru Mizushima.

**Project administration:** Noboru Mizushima.

**Supervision:** Ikuko Koyama-Honda, Noboru Mizushima.

**Validation:** Yoshitaka Kurikawa, Ikuko Koyama-Honda.

**Visualization:** Yoshitaka Kurikawa.

**Writing – original draft:** Yoshitaka Kurikawa, Ikuko Koyama-Honda, Noboru Mizushima.

**Writing – review & editing:** Yoshitaka Kurikawa, Ikuko Koyama-Honda, Norito Tamura, Seiichi Koike, Noboru Mizushima.

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
