## [Editor Report · Decision Letter 0]

5 Feb 2024

Dear Dr Mizushima, 

Thank you for submitting your manuscript via Review Commons entitled "Organelle landscape analysis using a multi-parametric particle-based method" for consideration as a Methods and Resources by PLOS Biology.

Your manuscript has now been evaluated by the PLOS Biology editorial staff as well as by an academic editor with relevant expertise and I am writing to let you know that we would like to invite you to submit a revision.

However, before we can send you the decision, we need you to complete your submission by providing the required metadata. To this end, please login to Editorial Manager where you will find the paper in the 'Submissions Needing Revisions' folder on your homepage. Please click 'Revise Submission' from the Action Links and complete all additional questions in the submission questionnaire.

Once your full submission is complete, your paper will undergo a series of checks. After your manuscript has passed the checks, we will send you the decision. To provide the metadata for your submission, please Login to Editorial Manager (https://www.editorialmanager.com/pbiology) within two working days, i.e. by Feb 07 2024 11:59PM.

Kind regards,

Ines

--

Ines Alvarez-Garcia, PhD

Senior Editor

PLOS Biology

---

## [Editor Report · Decision Letter 1]

8 Feb 2024

Dear Dr Mizushima,

Thank you for the submission via Review Commons of your manuscript entitled "Organelle landscape analysis using a multi-parametric particle-based method" and currently under consideration at PLOS Biology.

After careful evaluation of the manuscript, reviews and revision plan, we would like to invite you to revise the work to thoroughly address the reviewers' reports.

Your revised manuscript is likely to be sent for further evaluation by all or a subset of the reviewers.

**IMPORTANT - SUBMITTING YOUR REVISION**

3. Resubmission Checklist

a) *PLOS Data Policy*

b) *Published Peer Review*

d) *Blurb*

Please also provide a blurb which (if accepted) will be included in our weekly and monthly Electronic Table of Contents, sent out to readers of PLOS Biology, and may be used to promote your article in social media. The blurb should be about 30-40 words long and is subject to editorial changes. It should, without exaggeration, entice people to read your manuscript. It should not be redundant with the title and should not contain acronyms or abbreviations. For examples, view our author guidelines: https://journals.plos.org/plosbiology/s/revising-your-manuscript#loc-blurb

Sincerely,

Ines

--

Ines Alvarez-Garcia, PhD

Senior Editor

PLOS Biology

---

## [Decision Letter · Decision Letter 2]

24 Jun 2024

Dear Dr Mizushima,

Thank you for your patience while we considered your revised manuscript entitled "Organelle landscape analysis using a multi-parametric particle-based method" for publication as a Methods and Resources at PLOS Biology. This revised version of your manuscript has been evaluated by the PLOS Biology editors, the Academic Editor and two of the original reviewers.

Based on the reviews, we are likely to accept this manuscript for publication, provided you satisfactorily address the data and other policy-related requests stated below.

We expect to receive your revised manuscript within two weeks. 

*Published Peer Review History*

*Press*

Sincerely,

Ines

--

Ines Alvarez-Garcia, PhD

Senior Editor

PLOS Biology

DATA POLICY:

Thank you for providing the individual quantitative observations that underlie the data summarized in the figures and results of your paper. I have checked the files and I am missing the data from the following figures - if no required, please explain why:

Fig. 4E; Fig. 5C, D; Fig. S2A; Fig. S8A, B; Fig. 9A, B and Fig. S10A, B

Please also ensure that figure legends in your manuscript include information on WHERE THE UNDERLYING DATA CAN BE FOUND.

CODE POLICY

Reviewers’ comments

Rev. 1:

All my concerns have been properly addressed. Again, this would be a great technique for dynamic interactions of organelles.

Rev. 3:

The authors responded appropriately to all my comments, suggestions, and questions. I now recommend the manuscript for publication in PLOS Biology.

---

## [Editor Report · Decision Letter 3]

30 Jul 2024

Dear Dr Mizushima,

Thank you for the submission of your revised Methods and Resources entitled "Organelle landscape analysis using a multi-parametric particle-based method" for publication in PLOS Biology. On behalf of my colleagues and the Academic Editor, Ana Garcia-Saez, I am delighted to let you know that we can in principle accept your manuscript for publication, provided you address any remaining formatting and reporting issues. These will be detailed in an email you should receive within 2-3 business days from our colleagues in the journal operations team; no action is required from you until then. Please note that we will not be able to formally accept your manuscript and schedule it for publication until you have completed any requested changes.

PRESS

Sincerely, 

Ines

--

Ines Alvarez-Garcia, PhD

Senior Editor

PLOS Biology
